materials science/nanotechnology

photocatalyst, doped ZnO, sol–gel, methyl orange, TPD-CO$_2$

**Author for correspondence:**
M. F. Kasim
e-mail: muhdfir@uitm.edu.my

This article has been edited by the Royal Society of Chemistry, including the commissioning, peer review process and editorial aspects up to the point of acceptance.

# Comparative study on photocatalytic activity of transition metals (Ag and Ni)-doped ZnO nanomaterials synthesized via sol–gel method

A. K. Azfar[1,2], M. F. Kasim[1,2], I. M. Lokman[1,2], H. A. Rafaie[3] and M. S. Mastuli[1,2]

[1]Centre for Nanomaterials Research, Institute of Science, Level 3 Block C, and
[2]School of Chemistry and Environment, Faculty of Applied Sciences, Universiti Teknologi MARA, 40450 Shah Alam, Selangor, Malaysia
[3]Faculty of Applied Sciences, Universiti Teknologi MARA Pahang, 26400 Bandar Tun Abdul Razak, Jengka, Pahang, Malaysia

MFK, 0000-0003-3217-1413

Ag and Ni/ZnO photocatalyst nanostructures were successfully synthesized by a sol–gel method. In this work, the photocatalyst sample was systematically studied based on several factors affecting the performance of photocatalyst, which are size, morphology, band gap, textural properties and the number of active sites presence on the surface of the nanocatalyst. X-ray diffraction revealed that Ag/ZnO nanomaterials experienced multiple phases, meanwhile for Ni/ZnO the phase of nanomaterials were pure and single phase for stoichiometry less than 5%. Field emission scanning electron microscope (FESEM) showed almost all of the synthesized nanomaterials possessed a mixture of nanorods and spherical-like shape morphology. The Ag/ZnO showed high photocatalytic activity, producing at least 14th trials of nanocatalyst reusability on degradation of methyl orange under UV irradiation. Interestingly, this phenomenon was not observed in larger surface area of Ni/ZnO nanomaterials which supposedly favour photocatalytic activity, but instead producing poor photocatalytic performance. The main reasons were studied and exposed by temperature-programmed desorption of carbon dioxide (TPD–CO$_2$) which showed that incorporation of Ag into ZnO lattice has enhanced the number of active sites on the surface of the nanocatalyst. Whereas incorporation of Ni in

ZnO has lowered the number of active sites with respect to undoped ZnO. Active sites measurement is effective and significant, providing opportunities in developing an intensive study as an additional factor.

## 1. Introduction

Photocatalysis refers to the process that uses light irradiation to activate catalyst in order to initiate chemical reaction. This process aids in degradation of organic pollutants on daily basis and is beneficial for wastewater treatment system. Industrial effluent produces dyes, such as textiles, ceramics, paints, and pulps and papers industries, which multiply the level of toxicity in water streams. Extensive studies on photocatalysis work have covered a range of aspects, including the introduction of photocatalyst, ways of enhancing photodegradation efficiency, essential variables associated with photocatalytic efficiency, as well as advancement and photocorrosion [1]. This large scope has a counterbalance on environmental security as photocatalysis offers a general solution to address the pressing water pollution issue [2]. Semiconductor photocatalysts, such as zinc oxide (ZnO), titanium dioxide ($TiO_2$) and iron(III) oxide ($Fe_2O_3$), have been widely applied for their functionality on photodegradation [1]. Oxide-based materials possess high photocatalytic activity, thus giving better photocatalytic efficiency [3]. ZnO possesses a large and direct band gap, thus requiring massive amount of energy of incident beam (photons) to generate photoexcitation. Some of the charge carriers cannot participate in photocatalytic activity due to recombination phenomenon that results in dissipation of absorbed energy in the form of lattice vibration (heat) and photon generation (light). The rapid recombination of photogenerated electron and hole pairs deteriorates the performance of ZnO as a photocatalyst. Hence, in order to overcome this barrier, efficient transportation and separation of charge carriers in photocatalyst need to be emphasized by introducing dopants as electron scavengers [4]. To be precise, doping contributes in three ways: (i) narrowing band gap and promoting adsorption, (ii) improving conductivity of ZnO and mobility of charge carriers and (iii) altering the conduction band (CB) position and valence band (VB) of ZnO. Doping with other element in ZnO semiconductors has attracted attention of many researchers due to the higher amount of charge trap, hence reducing bulk recombination, as well as separating photogenerated electrons and holes more efficiently [4–6].

Various methods have been applied to synthesize doped ZnO, such as sol–gel [7–9], hydrothermal [10–12], combustion [13–15] and chemical vapour deposition [16,17]. However, all of these methods except sol–gel require severe reaction conditions such as high temperature, sophisticated techniques, high purity of gas, adjustable gas flow rate, expensive raw materials and so on. Thus, in this work, sol–gel method has been chosen due to simple operation, mild conditions and excellent crystalline structure of particles [18].

Previous works have proven that the photocatalytic activity of ZnO is significantly affected by its morphology [19], crystal size [20], crystal orientation [21] and also optical properties [22]. But, discussion on active sites analysis is still vague and most of the literature provides only light amount of study [23–27]. Though, the latest study by Xiao *et al.* reported in detail that larger specific surface area and more catalytic active sites led to improved performance of photocatalytic activity [28]. Meanwhile Yuan *et al.* discussed that active species scavengers influenced photogenerated holes and •OH radical which play an important role in photocatalysis [29]. In regard to these conditions, it is very crucial to investigate the number of active sites present in the surface of nanomaterials. This is because the presence of greater number of active sites will promote formation of active radicals. The goal in this work accounted in the present paper aims at bridging the information between fundamental and application work which attempt on proving active sites as an additional factor that should be notified. Thus, in this work, the prepared photocatalyst were extensively studied in terms of size, morphology, band gap, textural properties and the number of active sites present on the surface of the nanocatalyst. The materials were characterized via X-ray diffraction (XRD), field emission scanning electron microscope (FESEM), Brunauer–Emmett–Teller (BET) surface analysis, temperature-programmed desorption of carbon dioxide (TPD-$CO_2$) and UV–Vis spectrophotometer. The investigation of photodegradation was performed under UV-light irradiation.

# 2. Experimental

## 2.1. Materials

Zinc acetate dihydrate was purchased from R&M chemicals with 99.5% purity. Silver(II) acetate and nickel(II) acetate were purchased from Aldrich with 99% purity. These starting materials were mixed with absolute ethanol AnapuR.

## 2.2. Synthesis of materials

Ag and Ni/ZnO nanoparticles were synthesized at different stoichiometry values ($x = 1\%$, 3%, 5%, 7% and 10%). Zinc acetate dihydrate and silver acetate/nickel(II) acetate were dissolved under absolute ethanol and was stirred for 2 h to gain a homogeneous mixture. Base (ammonium hydroxide) was added to increase the pH value to pH 9, and this was followed by a heating process at 80°C. The materials underwent slow drying process and grey precursors were obtained within 24 h. For comparison, a control sample (undoped ZnO) was prepared by mixing zinc acetate dihydrate with absolute ethanol and processed with the similar procedure as above. The precursors were annealed at 400°C for 3 h. Next, structural studies on crystallinity were carried out after the annealing process using XRD (PANanalytical) X'pert Pro powder diffraction equipment. The morphology of the materials was assessed under FESEM (JEOL JSM-7600F). The band gap study, which in detail depicts light absorption properties, was performed under reflectance (%R) mode using Perkin Elmer Lambda 950 UV–Vis-NIR Spectrophotometer. The surface area was assessed using BELSORP-mini II instrument from BEL Japan Inc. The specific surface areas of undoped, Ag and Ni/ZnO were plotted under BET plot. Measurement of active sites were determined using TPD-$CO_2$.

## 2.3. Photocatalytic activity

The photocatalytic activity on Ag and Ni/ZnO nanoparticles was measured by determining the decomposition of methyl orange on each interval at constant room temperature. The catalyst loading was 100 mg of Ag and Ni/ZnO catalyst, in a medium beaker containing 100 ml of methyl orange solution with 10 ppm as the initial concentration. The UV-light irradiation was turned on at 352 nm wavelength and 8 W. The dye solution was extracted out at every 40 min interval. The photocatalytic analysis was performed using UV–Vis spectrophotometer under absorbance, (A), mode. The methyl orange absorption peak was measured at 464 nm. Photodegradation efficiency (%) was measured in regard to the maximum photodegradation collected at each interval. Photodegradation rate constant, $k$, was calculated for all the samples. Absorption controlled graph was produced prior to photodegradation by excluding UV-light irradiation.

# 3. Results and discussion

## 3.1. Phase studies

Figure 1*a,b* illustrates the XRD pattern ranged between 20° and 90° for Ag and Ni/ZnO nanostructures, respectively. Good crystallinity was achieved as the diffracted peaks displayed good match with the ICDD reference no. 01-089-0510 of ZnO wurtzite hexagonal with a space group of P63mc. As for the Ag/ZnO (figure 1*a*), diffraction peaks of Ag metal were detected at (111), (200), (220) and (311) crystal plane in correlation with standard Ag ICDD reference no. 01-087-0717. This occurrence is attributed to the huge variance in terms of ionic size between $Ag^+$ ions and $Zn^{2+}$ ions in the ZnO system. With that being mentioned, the existence of Ag metal seemed to affect the fabrication of ZnO [30,31]. Based on the XRD pattern, no shifting was observed in the peak position for Ag/ZnO samples. This indicates that the existence of Ag particles were not slotted into the lattice, but squeezed in the grain boundaries of ZnO crystallites [30]. For Ni/ZnO, as shown in figure 1*b*, it was revealed that the materials have single phase of ZnO wurtzite hexagonal structure for the Ni content less than 5%. But as Ni content exceeds 5%, multiphases were observed corresponding to NiO and Ni metals at (200) and (111) plane with standard Ni ICDD reference nos. 01-073-1519 and 00-001-1266, respectively. Although the XRD peak position did not exhibit any changes, some alteration was noted for peak intensity and width. In this case, introduction of Ag/Ni did not move the peak to a lower position, as

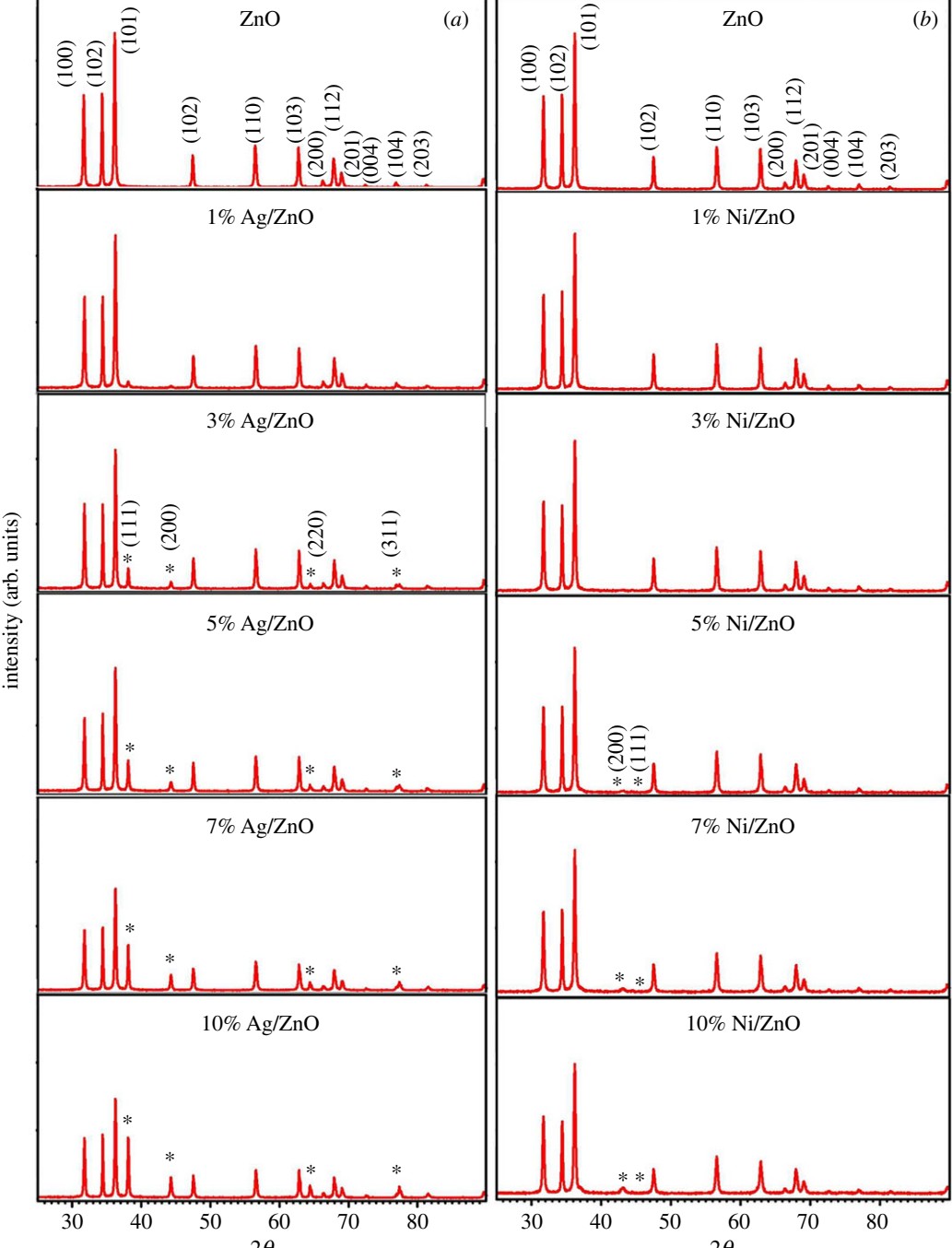

**Figure 1.** The XRD pattern of (*a*) various amount of Ag-doped ZnO and (*b*) various amount of Ni-doped ZnO materials.

reported by Mohammadzadeh *et al*. and Goswami & Sahai [32,33]. In fact, the impurities for peak intensity for Ag metal increased as the stoichiometry increased at peak position (111).

## 3.2. Morphology and elemental composition analysis

Figures 2 and 3 illustrate the morphology of Ag and Ni/ZnO with varied stoichiometry values, respectively. The morphology of Ag/ZnO (figure 2) reflected similarly as undoped ZnO nanorod-like-shaped structure in all stoichiometry. By contrast, the morphology above 3% Ni/ZnO (figure 3) exhibited a mixture of spherical and short nanorod-like-shaped structures. The morphology for materials for 10% Ni content in ZnO solely exhibited a spherical shape. Summarization of morphologies, crystallite dimensions (length and diameter) and aspect ratio (length over diameter) of

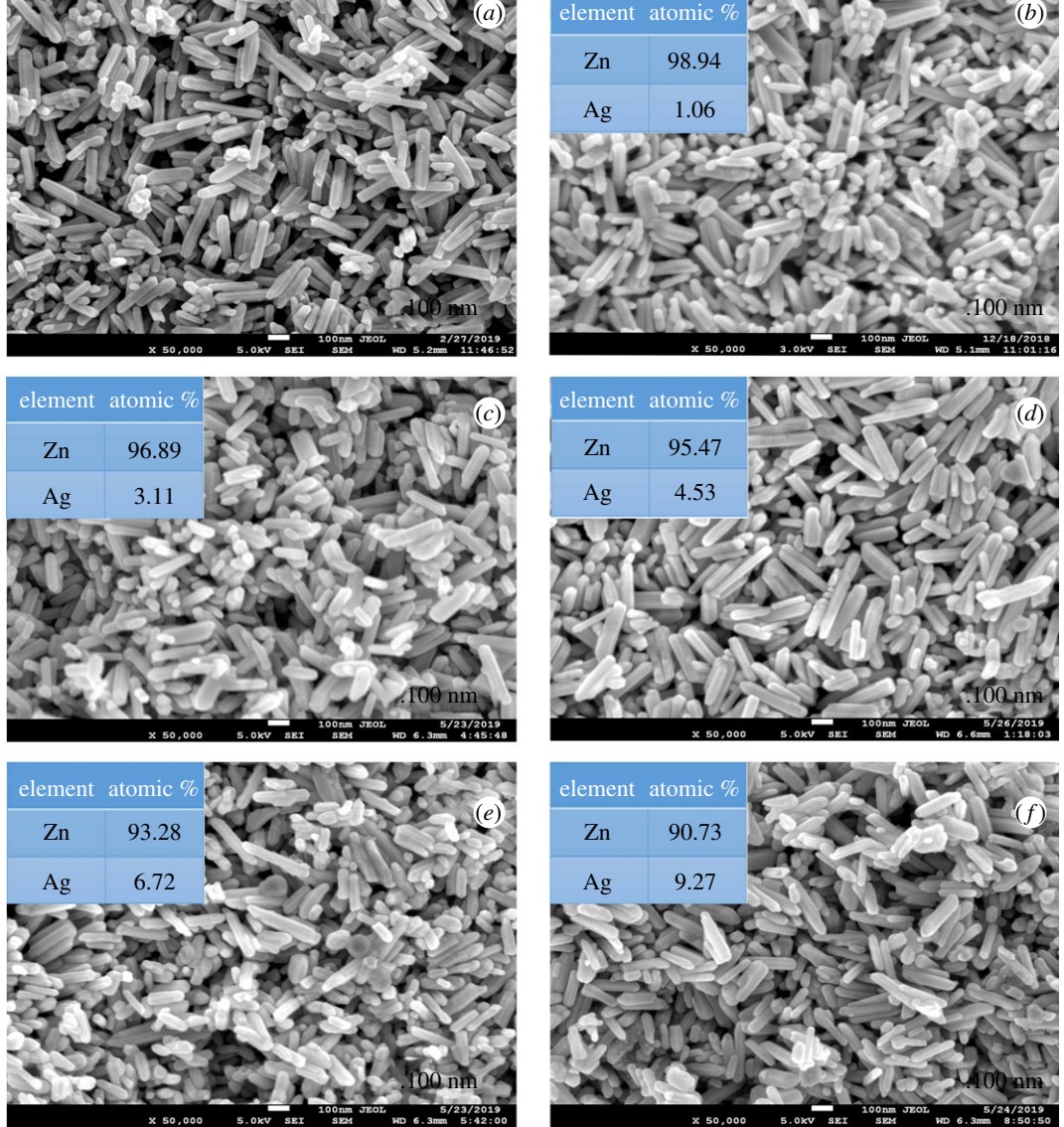

**Figure 2.** SEM images of (*a*) undoped ZnO, (*b*) 1% Ag/ZnO, (*c*) 3% Ag/ZnO, (*d*) 5% Ag/ZnO, (*e*) 7% Ag/ZnO and (*f*) 10% Ag/ZnO nanomaterials.

Ag and Ni/ZnO are tabulated in electronic supplementary material, table S1. By doping Ag into ZnO, it did not show much change in terms of morphology and crystal dimension as compared to undoped ZnO. By contrast, by doping Ni into ZnO (figure 3), the morphology started to become spherical-like shape as the Ni content increases and average length decreases, but only slight changes are noted in crystal's dimension when compared with undoped ZnO. For this reason, size and morphology does not take much role in affecting photocatalysis in this work due to its resemblance in both Ag and Ni/ZnO.

Further characterization on elemental composition was performed using EDS for Ag and Ni/ZnO nanomaterials (see electronic supplementary material, figures S1 and S2). The EDS results show (see inset figures 2 and 3) that all the synthesized material have good agreement with the stoichiometry calculation.

## 3.3. Band gap determination

Band gaps are evaluated by plotting Tauc plot from the absorption edges of the reflectance spectra shown in figures 4*a* and 5*a* for Ag and Ni/ZnO respectively. The Tauc relation was applied via equation below

$$(\alpha h v) = C(h v - Eg)^{n}. \tag{3.1}$$

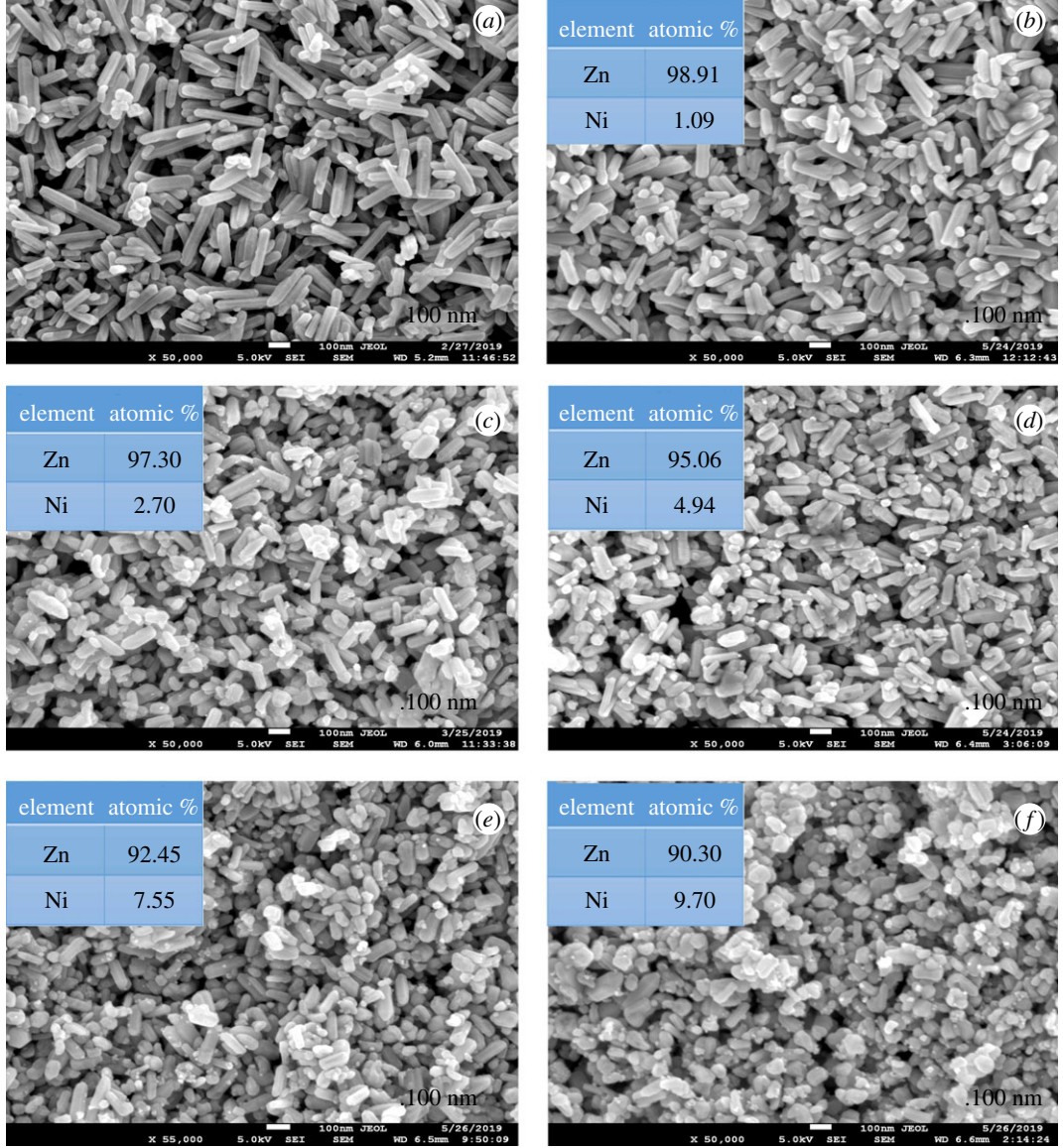

| element | atomic % |
|---------|----------|
| Zn | 98.91 |
| Ni | 1.09 |

| element | atomic % |
|---------|----------|
| Zn | 97.30 |
| Ni | 2.70 |

| element | atomic % |
|---------|----------|
| Zn | 95.06 |
| Ni | 4.94 |

| element | atomic % |
|---------|----------|
| Zn | 92.45 |
| Ni | 7.55 |

| element | atomic % |
|---------|----------|
| Zn | 90.30 |
| Ni | 9.70 |

**Figure 3.** SEM images of (*a*) undoped ZnO, (*b*) 1% Ni/ZnO, (*c*) 3% Ni/ZnO, (*d*) 5% Ni/ZnO, (*e*) 7% Ni/ZnO and (*f*) 10% Ni/ZnO nanomaterials.

From the Tauc relation, $\alpha$ represents the absorption coefficient of the material, h denotes Planck's constant, $v$ reflects the frequency of light, $C$ is the proportionality constant, $E_g$ refers to the band gap energy and $n = \frac{1}{2}$ (for direct transition mode materials), since ZnO is classified under direct band gap semiconductor [34,35]. The absorption coefficient in this study was determined by

$$\alpha = k \ln\left(\frac{R_{max} - R_{min}}{R - R_{min}}\right). \tag{3.2}$$

Based on the absorption coefficient, $k$ represents a constant, $R_{max}$ stands for the maximum reflectance and $R_{min}$ refers to the minimum reflectance. Equations (3.1) and (3.2) produce the following:

$$(\alpha h v)^2 = C' (h v - E_g). \tag{3.3}$$

Extrapolation was derived from the graph of $(\alpha h v)^2$ against h$v$, while band gap energy was determined once it met the line of $x$-axis, abscissa. The Tauc plot graph and the extrapolation are illustrated in figures 4*b* and 5*b* for Ag and Ni/ZnO, respectively. The band gap values are tabulated in electronic supplementary material, table S1. It was revealed that the band gap of Ag/ZnO did not consistently change with increment of Ag content, and this is happened because Ag⁺ ions do not take

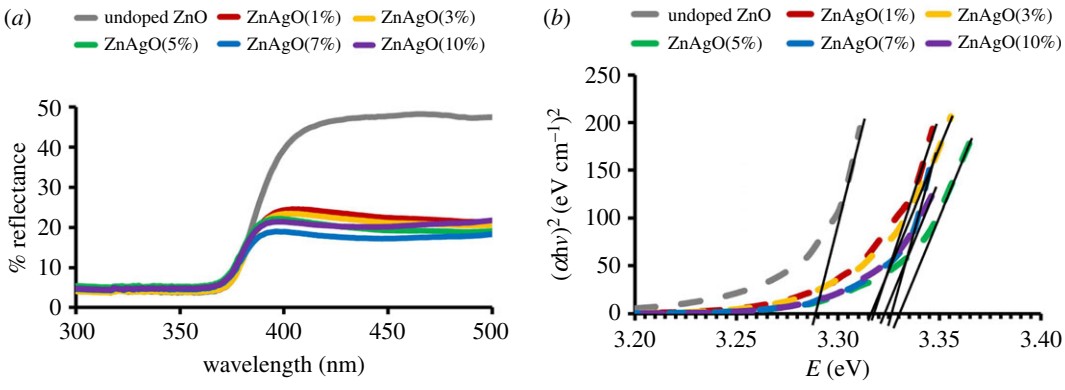

**Figure 4.** The result of (*a*) UV–visible spectra and (*b*) Tauc plots of Ag/ZnO with different stoichiometry.

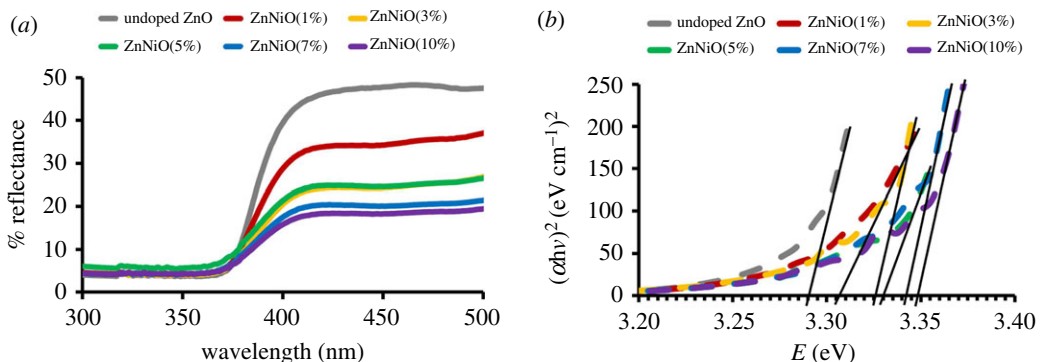

**Figure 5.** The result of (*a*) UV–visible spectra and (*b*) Tauc plots of Ni/ZnO with different stoichiometry.

the place of the $Zn^{2+}$ ions in the lattice crystal, which means that $Ag^+$ ions do not contribute in the VB of ZnO materials. It is believed that the $Ag^+$ ions only existed on the surface of the ZnO materials. This works well-agreed with other researchers [5,36–38]. By contrast, for the Ni/ZnO, the $E_g$ value increased with increment of Ni content. The change in band gap happened due to the substitution of $Ni^{2+}$ on $Zn^{2+}$ site in the crystal lattice. The electrons of $Ni^{2+}$ ions are contributed in the valence region of the ZnO and thus lead to the changes in band gap. The increased Ni content in the ZnO system led to the presence of NiO phase due to incomplete substitution. This situation weakened the bonding of Zn-O, but induced the growth of Ni-O. The changes noted in band gap values for Ag- and Ni-doped ZnO are attributable to grain size, structural parameter and carrier concentration [33,39]. Band gap has a significant role in determining the performance of photocatalysis. This work proves that other factors, as mentioned, may topple band gap as the main factor for photodegradation of methyl orange, as elaborated in the following section.

## 3.4. Surface area analysis

Electronic supplementary material, figures S3 and S4 illustrate $N_2$ adsorption–desorption and pore size distribution on undoped, Ag and Ni/ZnO. All the adsorption isotherms can be classified as Type II based on the classification standard of International Union of Pure and Applied Chemistry (IUPAC). The porosity under Type II criteria displays its macroporous nature. The specific surface area, the total pore volume and the average pore diameter of undoped, Ag and Ni/ZnO were determined by constructing BET plot (see electronic supplementary material, figures S5 and S6). The textural properties for all samples are tabulated in electronic supplementary material, table S1. It was found that Ni/ZnO samples have the largest specific surface area, followed by undoped and Ag/ZnO. By having larger specific surface area, it will give an advantage in the performance of photocatalytic activity. Nevertheless, Ni/ZnO failed to generate better photocatalytic activity, when compared with either undoped or Ag/ZnO. In this case, it is believed that the sites of Ni/ZnO are less active or inactive towards photocatalytic reaction [40,41].

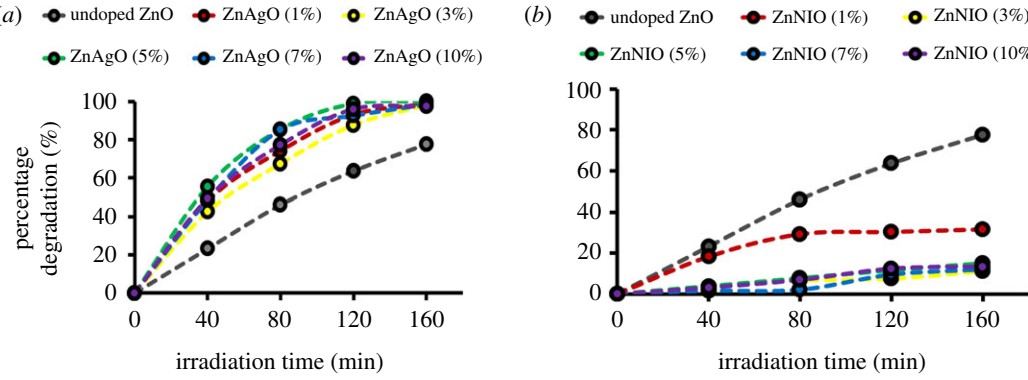

**Figure 6.** Photocatalytic degradation efficiency on different stoichiometry of (a) Ag/ZnO and (b) Ni/ZnO nanomaterials.

## 3.5. Photocatalytic activity

The photodegradation of undoped, Ag and Ni/ZnO nanomaterials had been determined by assessing each interval (40 min) of the degradation of 10 ppm methyl orange aqueous solution under 352 nm 8 W UV light irradiation. Electronic supplementary material, figures S7(a) and S7(b) portray that photocatalysis only occurred with the presence of light as the primary source and catalyst. This means no photodegradation would occur without the presence of light and catalyst, resulting in nil changes on the absorption spectra. Electronic supplementary material, figures S8 and S9 show the time-dependent absorbance spectra for Ag and Ni/ZnO from 350 to 550 nm for 160 min with the presence of UV light irradiation. Based on the plot, the characteristic peak, $\lambda_{\max}$, for methyl orange was at 464 nm. All the stoichiometry of Ag/ZnO displayed promising photodegradation with total clarity after 160 min, while Ni/ZnO generated low photocatalytic efficiency, wherein 1% gave the best output amongst other stoichiometry.

From the graph, photodegradation efficiency (%) was evaluated based on the measured absorbance from each interval. The photocatalytic degradation efficiency (%) is expressed in equation (3.4), as follows:

$$\left[\frac{(C_0 - C)}{C_0}\right] \times 100 = \left[\left(\frac{A_0 - A}{A_0}\right)\right] \times 100. \tag{3.4}$$

From the equation, $C_0$ represents the initial dye concentration, while $C$ denotes the dye concentration on each interval in terms of time (minutes). Meanwhile, $A_0$ stands for the initial absorbance, whereas $A$ refers to the absorbance on each interval at specific absorption wavelength of methyl orange, which is at 464 nm [42,43].

Figure 6a,b shows the photocatalytic activity of Ag and Ni/ZnO, respectively. It was found that 5% of Ag content in ZnO resulted in the best degradation efficiency with 99.93%, when compared with 1%, 3%, 7% and 10%, which gave 98.65%, 98.00%, 98.59% and 97.47% degradation efficiency, respectively. Meanwhile, low photocatalytic degradation efficiency was recorded for Ni/ZnO with only 1% of Ni content in ZnO giving the best outcomes amongst the respective stoichiometry with 31.68%. No appreciable photodegradation efficiency was observed for 3%, 5%, 7% and 10% with 11.12%, 15.07%, 12.11% and 13.44%, respectively.

The photodegradation rate constant, $k$, of Ag and Ni/ZnO had been assessed using the kinetic model suggested by Langmuir–Hinshelwood, which is pseudo-first kinetics model of photocatalysis [44], as given in equation (3.5) below:

$$\ln\left(\frac{C}{C_0}\right) = -kt, \tag{3.5}$$

From the equation, $C_0$ represents the initial concentration of methyl orange, $C$ refers to the concentration of methyl orange on each interval and $t$ is the irradiation time. The graph of ln ($C/C_0$) against $t$ exhibits a linear relationship in figure 7a,b, and followed by pseudo-first-order reaction kinetics. The pseudo-first-order rate constant, $k$, and linear regression, $R^2$, are tabulated in electronic supplementary material, table S1 for Ag and Ni/ZnO, respectively. Five per cent of Ag content and 1% of Ni content in ZnO gave the highest $k$ values amongst their respective stoichiometry. These results show that

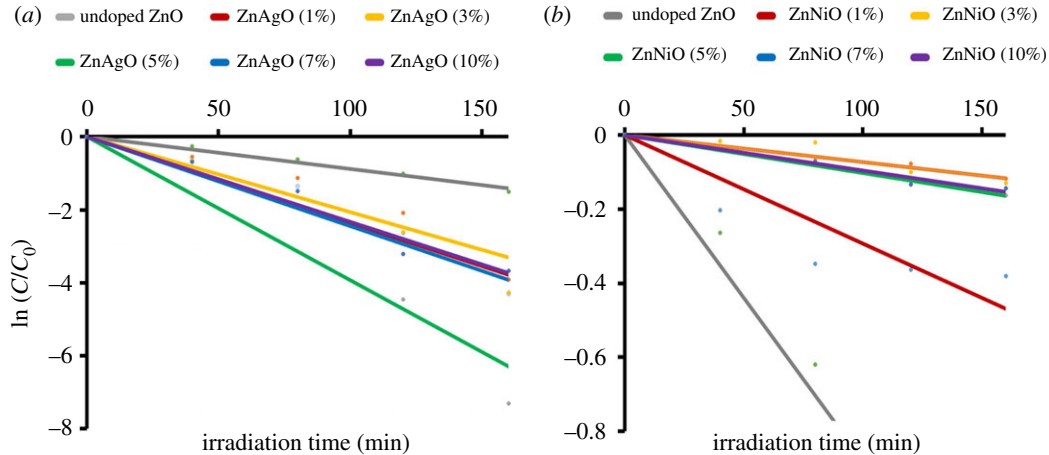

**Figure 7.** The curves of ln ($C/C_0$) versus time for photodegradation of methyl orange on different stoichiometry of (*a*) Ag/ZnO and (*b*) Ni/ZnO nanomaterials.

**Table 1.** Basic sites of Ag- and Ni/ZnO determined by TPD-$CO_2$.

| sample | temperature (°C) | amount of $CO_2$ desorbed ($\mu$mol g$^{-1}$) | total amount of basicity ($\mu$mol g$^{-1}$) |
|---|---|---|---|
| undoped ZnO | 465 | 57.92 | 119.77 |
|  | 754 | 61.85 |  |
| 10% Ag/ZnO | 140 | 19.35 | 166.53 |
|  | 494 | 147.18 |  |
| 10% Ni/ZnO | 428 | 34.08 | 61.38 |
|  | 599 | 27.30 |  |

doping Ag into ZnO improved the photocatalytic performance with respect to undoped ZnO. By contrast, doping Ni into ZnO caused detrimental effects to the photocatalytic performance. Even though BET results (see electronic supplementary material, table S1) showed that Ni/ZnO has larger specific surface area over undoped and Ag/ZnO, which supposedly give an advantage to the photocatalytic performance, but yet it turned out the photocatalytic test in poor performance. Since there is not much difference in crystal dimensions and band gap among the materials, it is believed that these factors do not play a vital role in the performance of photocatalyst nanomaterials.

## 3.6. Active sites measurement

Further characterization on the number of active sites present on the surface of the materials are carried out via temperature-programmed desorption of carbon dioxide (TPD-$CO_2$). TPD profiles portrayed the interaction between $CO_2$ molecules and photocatalyst surfaces tabulated in table 1. Desorption of $CO_2$ happened at temperatures ranging between 300°C and 550°C, which shows all samples consist of high basic centres. Both undoped ZnO and Ag/ZnO showed strong desorption peak at temperature above 350°C. Ag/ZnO possessed the highest total amount of basicity (figure 8), which gave the highest active sites on photocatalyst surface. Therefore, Ag/ZnO is a likely-looking spot to initiate a better photocatalytic performance compared to Ni/ZnO. Asymmetrical, weak and decreasing in desorption peak was noted on Ni/ZnO may be due to diffusion limitation and decompostion of carbonates species. Diffusion limitation caused by narrow pore size distribution of dopant Ni resulted in low total amount of active sites. As being related to surface area analysis, though the pore size of both Ag and Ni/ZnO (see electronic supplementary material, table S1) having a slight resemblance, the sites in Ni are believed to be less active. Therefore, the coverage of reactive adsorbed species on photocatalyst surfaces will be less, thus leading to inefficient photocatalytic performance [45–47].

Photocatalytic activity for Ni/ZnO displayed that 1% of Ni content gave the highest photocatalytic performance, when compared with other stoichiometry in Ni classes. Technically, increment in the stoichiometry of Ni content resulted in lower photocatalytic activity. Interestingly, this case does not

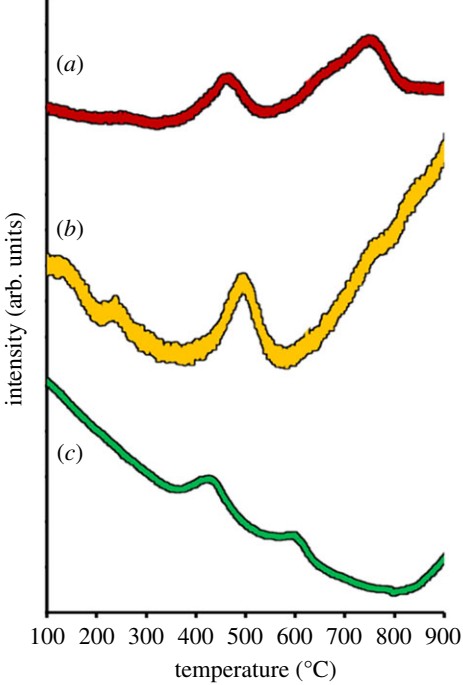

**Figure 8.** $CO_2$-TPD profiles of (a) undoped ZnO, (b) 10% Ag/ZnO and (c) 10% Ni/ZnO nanomaterials.

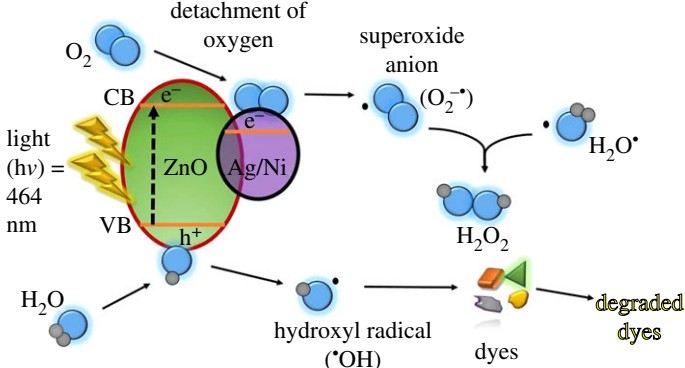

**Figure 9.** Schematic band diagram on photocatalytic reaction.

apply on Ag/ZnO nanomaterials, as all of its stoichiometry turned out to give a good development on photocatalysis. Comparison between the highest content of Ag and Ni/ZnO suggests that active sites control photocatalytic activity in this work, and this has been proven through TPD-$CO_2$ result in table 1. The TPD-$CO_2$ result answered the observed photocatalytic behaviour of both doped samples that greater number of active sites will enhance the photocatalytic activity by promoting more formation of active oxidant species [45,48].

## 3.7. Mechanism of photocatalysis

Basically, the process of photocatalysis (figure 9) takes place when energy is higher than source, in comparison to the energy gap of a semiconductor, which is ZnO in this study. Thus, electrons excited from VB jump to CB to form positively hole and electron on the surface of ZnO (equation (3.6)). These electron-hole pairs involved in redox reaction as shown in equations (3.7) and (3.8) producing hydroxyl radicals. The dopants (Ag and Ni) serve as electron scavenger (equations (3.9) and (3.10)) to trap excited electron that intercepts the recombination between the pair of photogenerated holes, thus resulting in increased life span of the excited electron. This gives more time for both excited electron and holes to react with $H_2O$ and generate exceptional oxidant species ($\bullet O_2^-$ and $H_2O_2$), which later degrades the dyes (equations (3.11)–(3.14)). The oxidant species is non-selective and is highly reactive

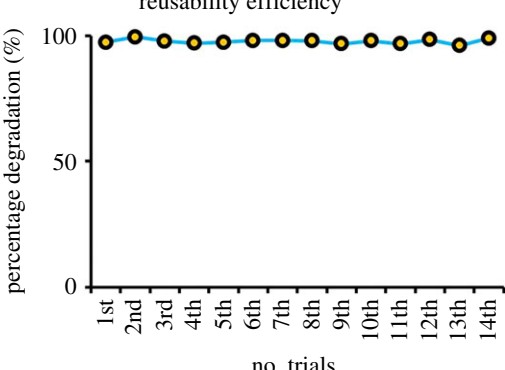

**Figure 10.** The graph of reusability efficiency of 5% Ag/ZnO nanomaterial.

that has the ability to degrade pollutants by breaking down organic bonds, including destroying the aromatic rings of dye molecules [3,49]. Generating more oxidant species enhances the photocatalytic performance. The possible mechanism of photocatalytic activity of Ag and Ni/ZnO are proposed as follows:

$$ZnO + h\nu \rightarrow ZnO\ (e_{CB}^- + h_{VB}^-), \tag{3.6}$$

$$ZnO(h_{VB}^+) + H_2O \rightarrow ZnO + H^+ + \bullet OH, \tag{3.7}$$

$$ZnO(h_{VB}^+) + OH^- \rightarrow ZnO + \bullet OH, \tag{3.8}$$

$$(e_{CB}^-) + (Ag^+, Ni^{2+}) \rightarrow (Ag, Ni^+), \tag{3.9}$$

$$(Ag, Ni^+) + O_2 \rightarrow (Ag^+, Ni^{2+}) + \bullet O_2^-, \tag{3.10}$$

$$\bullet O_2^- + H_{aq}^+ \rightarrow HO_2 \bullet, \tag{3.11}$$

$$HO_2 \bullet + HO_2 \bullet \rightarrow H_2O_2 + O_2, \tag{3.12}$$

$$H_2O_2 + e_{CB}^- \rightarrow \bullet OH + OH^- \tag{3.13}$$

and

$$\bullet OH/h_{VB}^- + MO\ dye \rightarrow degradation\ products + CO_2 + H_2O. \tag{3.14}$$

## 3.8. Photostability and reusability

Reusability of the catalyst is one of the advantages for photocatalysis system [50]. This work produced an experimental continuation for its catalyst reusability. Five per cent Ag/ZnO has been tested and displayed a prominent character for photocatalysis, which suits the advantage of semiconductors. Results in this work are tabulated in table 2, conjoint with previous similar works on photocatalytic performance-wise. Figure 10 illustrates the reusability on 5% Ag content that had undergone 14 consecutive attempts without removing the catalyst powder batches upon batches. The degraded methyl orange was discarded and replaced with new but similar initial methyl orange batch at 10 ppm. The Ag/ZnO appears to be completely stable and could hardly deactivate even after the 14th attempt. Therefore, this material can be further conducted and used umpteenth trials due to absence of inhibitors and poison mainly from physical and chemical reactions.

# 4. Conclusion

All in all, photocatalysis operation and performance conclude fundamental and application work. Synthesis using modified sol–gel method was performed on Ag and Ni/ZnO nanomaterials from varying stoichiometry values ($x = 1\%$, 3%, 5%, 7% and 10%). This paper disclosed that only Ag/ZnO exerts greater performance on photocatalysis, but this does not apply to Ni/ZnO, which has larger surface area. The photocatalytic performance showed that 5% of Ag content in ZnO exhibited the most degradation at 99.93%, which refers to satisfactory outcome in total degradation. To date, without a doubt, factors such as band gap, surface area and crystallite size have been proven thoroughly in influencing photocatalytic activity. However, active sites measurement could arguably be an additional factor and might be a greater choice in substituting photocatalysis features. This work suggested that active sites of catalyst are significant in affecting photocatalytic activity as Ni/ZnO is believed to be less active than Ag/ZnO. Hence, active sites measurement should be

**Table 2.** Overview of published data on Ag and Ni/ZnO nanostructures and photocatalytic performances for azo-dyes (methyl orange (MO) and methylene blue (MB)).

| catalyst | dye | initial concentration of model pollutant (mg l$^{-1}$) | catalyst loading (mg) | experimental conditions | duration removal (min) | percentage degradation (%) | reusability times of catalyst | refs |
|---|---|---|---|---|---|---|---|---|
| 4.6% Ag/ZnO | MB | 10 | 10 | UV lamp = 6 W, λ = 254 nm | 60 | 74.38 | n.a. | [3] |
| 4% Ni/ZnO | MB | 20 | 50 | UV lamp = 15 W, λ = 365 nm | 360 | 46.00 | n.a. | [51] |
| 0.5% Ag/ZnO | MB | $10^{-5}$ | 50 | fluorescence lamp = 60 W | 180 | 98.00 | n.a. | [32] |
| 7% Ni/ZnO | MB | 5 | 40 | Hg lamp = 300 W, λ = 280 nm – 400 nm | 200 | 83.00 | n.a. | [52] |
| 20% Ag/ZnO | MO | 10 | 10 | UV lamp = 10 W, λ = 365 nm | 120 | 88.00 | n.a. | [53] |
| 2% Al/4% Ni/ZnO | MO | 10 | 50 | halogen lamp = 100 W, λ > 450 nm | 105 | 66.00 | 5th cycle | [54] |
| 5% Ag/ZnO | MO | 20 | 1000 | UV lamp = 300 W, λ = 250 nm – 365 nm | 60 | 100.00 | 5th cycle | [55] |
| ZnFe$_2$O$_4$/ZnO/Ag | MO | 10 | 50 | fluorescence lamp = 85 W, λ = 430 nm – 630 nm | 420 | 84.00 | n.a. | [56] |
| 10% Ag/ZnO | MO | 20 | 1000 | UV lamp = 300 W, λ = 250 nm – 365 nm | 30 | 100.00 | 2nd cycle | [57] |
| Ag/ZnO | MO | 10 | 75 | high pressure Hg lamp = 250 W, λ = 365 nm | 50 | 98.40 | 5th cycle | [58] |
| Ag/ZnO | MO | 10 | 300 | UV lamp = 40 W, λ = 365 nm | 120 | 90.00 | 5th cycle | [59] |
| 5% Ag/ZnO 1% Ni/ZnO | MO | 10 | 100 | UV lamp = 8 W, λ = 352 nm | 160 | 99.93 31.68 | 14th cycle | this work |

intensively studied and included in photocatalysis work. Catalyst reusability on 5% Ag/ZnO showed an amazing outcome with 14 cycles done with no negative effect on the performance. Therefore, Ag/ZnO which has been well researched can undergo series of larger wastewater treatment system in real work due to its photostability feature.

Data accessibility. All datasets and codes have been uploaded in the Dryad Digital Repository and have been made publicly available: https://dx.doi.org/10.5061/dryad.djh9w0vvs [60].

Authors' contributions. M.F.K. conceived the ideas and designed the experiments. A.K.A. performed the experiments, acquired, analysed and interpreted data with M.F.K. H.A.R. co-wrote the manuscript with M.F.K. M.S.M analysed and interpreted data on surface area and active sites measurement with I.M.L. All authors discussed the results and commented on or revised the manuscript.

Competing interests. We have no competing interests.

Funding. This research is funded by Universiti Teknologi MARA, Malaysia, under the UiTM Research grant no. 600-IRMI/DANA KCM 5/3/LESTARI (103/2017).

Acknowledgements. The authors wish to thank the internal grant agency funded by Universiti Teknologi MARA, Malaysia, under the UiTM Research grant no. 600-IRMI/DANA KCM 5/3/LESTARI (103/2017). Special gratitude to Institute of Science, UiTM for provision of research instruments.

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
