## [Reviewer comments · Royal Society Open Science]

Review History

RSOS-191590.R0 (Original submission)

Review form: Reviewer 1

Is the manuscript scientifically sound in its present form?

Yes

Are the interpretations and conclusions justified by the results?

Yes

Is the language acceptable?

Yes

Do you have any ethical concerns with this paper?

No

Have you any concerns about statistical analyses in this paper?

No

Recommendation?

Major revision is needed (please make suggestions in comments)

Comments to the Author(s)

The authors prepared Ag- and Ni/ZnO photocatalyst nanostructures, and studied their photocatalytic activity. The authors did a series of characterizations for the modified material. This work is somewhat interesting and worthy to be published in Royal Society Open Science; however, the following comments need to be addressed:

1. The title is inappropriate: is it '(Ag- and Ni-doped ZnO)' ?
2. From the introduction, the significance of the work is insufficiency. There are many strategies to enhance the photocatalytic activity of ZnO. Particularly, the latest references about the ZnO or its hybrids are suggested to be added for the real value of general readers, as recommended below: Journal of physics and chemistry of solids, 106, (2017): 1-9; Journal of physics d-applied physics, 51(7), (2018): 075501. Thus, it can be clearly and specifically pointed out the novelty of this work in the introduction.
3. The quality of figures is too poor. Figures 1 and 2 should be merged. In Figures 3 and 4, the atomic % is misunderstanding: 0% O element? Figures 5-9 need to be revised.
4. The mechanism of photocatalytic activity of Ag- and Ni/ZnO nanomaterials should be discussed.
5. The whole manuscript (especially abstract and introduction) needs to be improved.
6. In this manuscript, a large number of grammar errors may be found, and some expressions used in manuscript are confusing. The authors should polish English carefully.

Review form: Reviewer 2

Is the manuscript scientifically sound in its present form?

Yes

Are the interpretations and conclusions justified by the results?

No

Is the language acceptable?

Yes

Do you have any ethical concerns with this paper?

No

Have you any concerns about statistical analyses in this paper?

No

Recommendation?

Major revision is needed (please make suggestions in comments)

Comments to the Author(s)

Authors prepared the Ag- and Ni/ZnO photocatalyst nanostructures by a sol-gel method and investigated the effects of size, morphology, band gap, surface area and the number of active sites on the properties of photocatalyst. However, some issues exist in paper. I suggested revising it before being published.

- In paper, there are little explanations of reason. For example, there are the effect of size, morphology, etc. Why?

- Draft writing should be formatted and meet the requirement of journal. For example, paper abstract and characterization are required.
- labels unit at x-/y axis should be marked in Figure 1, 2,....., the font size in Figure 9 is not consistent with other Figures
- Please check the spelling of words.

Decision letter (RSOS-191590.R0)

28-Oct-2019

Dear Dr Kasim:

Title: Comparative study on photocatalytic activity of transition metals (Ag- and Ni-doped ZnO) nanomaterials synthesised via sol-gel method
Manuscript ID: RSOS-191590

The editor assigned to your manuscript has now received comments from reviewers. We would like you to revise your paper in accordance with the referee and Subject Editor suggestions which can be found below (not including confidential reports to the Editor). Please note this decision does not guarantee eventual acceptance.

Please submit your revised paper before 20-Nov-2019. Please note that the revision deadline will expire at 00.00am on this date. If we do not hear from you within this time then it will be assumed that the paper has been withdrawn. In exceptional circumstances, extensions may be possible if agreed with the Editorial Office in advance. We do not allow multiple rounds of revision so we urge you to make every effort to fully address all of the comments at this stage. If deemed necessary by the Editors, your manuscript will be sent back to one or more of the original reviewers for assessment. If the original reviewers are not available we may invite new reviewers.

RSC Associate Editor:
 Comments to the Author:
 (There are no comments.)

RSC Subject Editor:
 Comments to the Author:
 (There are no comments.)

Reviewers' Comments to Author:
 Reviewer: 1

Comments to the Author(s)

The authors prepared Ag- and Ni/ZnO photocatalyst nanostructures, and studied their photocatalytic activity. The authors did a series of characterizations for the modified material. This work is somewhat interesting and worthy to be published in Royal Society Open Science; however, the following comments need to be addressed:

1. The title is inappropriate: is it '(Ag- and Ni-doped ZnO)' ?
2. From the introduction, the significance of the work is insufficiency. There are many strategies to enhance the photocatalytic activity of ZnO. Particularly, the latest references about the ZnO or its hybrids are suggested to be added for the real value of general readers, as recommended below: Journal of physics and chemistry of solids, 106, (2017): 1-9; Journal of physics d-applied physics, 51(7), (2018): 075501. Thus, it can be clearly and specifically pointed out the novelty of this work in the introduction.
3. The quality of figures is too poor. Figures 1 and 2 should be merged. In Figures 3 and 4, the atomic % is misunderstanding: 0% O element? Figures 5-9 need to be revised.
4. The mechanism of photocatalytic activity of Ag- and Ni/ZnO nanomaterials should be discussed.
5. The whole manuscript (especially abstract and introduction) needs to be improved.
6. In this manuscript, a large number of grammar errors may be found, and some expressions used in manuscript are confusing. The authors should polish English carefully.

Reviewer: 2

Comments to the Author(s)

Authors prepared the Ag- and Ni/ZnO photocatalyst nanostructures by a sol-gel method and investigated the effects of size, morphology, band gap, surface area and the number of active sites on the properties of photocatalyst. However, some issues exist in paper. I suggested revising it before being published.

- In paper, there are little explanations of reason. For example, there are the effect of size, morphology, etc. Why?

- Draft writing should be formatted and meet the requirement of journal. For example, paper abstract and characterization are required.
- labels unit at x-/-y axis should be marked in Figure 1, 2....., the font size in Figure 9 is not consistent with other Figures
- Please check the spelling of words.

Author's Response to Decision Letter for (RSOS-191590.R0)

See Appendices A & B.

RSOS-191590.R1 (Revision)

Review form: Reviewer 1

Is the manuscript scientifically sound in its present form?

Yes

Are the interpretations and conclusions justified by the results?

Yes

Is the language acceptable?

Yes

Do you have any ethical concerns with this paper?

No

Have you any concerns about statistical analyses in this paper?

No

Recommendation?

Accept as is

Comments to the Author(s)

The authors have provided satisfactory modification to the manuscript by incorporating the raised comments. I appreciate authors effort and recommend it for publication.

Review form: Reviewer 2

Is the manuscript scientifically sound in its present form?

Yes

Are the interpretations and conclusions justified by the results?

Yes

Is the language acceptable?

Yes

Do you have any ethical concerns with this paper?

No

Have you any concerns about statistical analyses in this paper?

No

Recommendation?

Accept with minor revision (please list in comments)

Comments to the Author(s)

Paper has been improved after major revision. I recommend it to be published in journal . I suggest to remove some results to supporting material, for example, Table1. please consider it.

Decision letter (RSOS-191590.R1)

13-Jan-2020

Dear Dr Kasim:

Title: Comparative study on photocatalytic activity of transition metals (Ag and Ni doped ZnO) nanomaterials synthesised via sol-gel method

Manuscript ID: RSOS-191590.R1

Thank you for submitting the above manuscript to Royal Society Open Science. On behalf of the Editors and the Royal Society of Chemistry, I am pleased to inform you that your manuscript will be accepted for publication in Royal Society Open Science subject to minor revision in accordance with the referee suggestions. Please find the reviewers' comments at the end of this email.

The reviewers and handling editors have recommended publication, but also suggest some minor revisions to your manuscript. Therefore, I invite you to respond to the comments and revise your manuscript.

Because the schedule for publication is very tight, it is a condition of publication that you submit the revised version of your manuscript before 22-Jan-2020. Please note that the revision deadline will expire at 00.00am on this date. If you do not think you will be able to meet this date please let me know immediately.

1) A text file of the manuscript (tex, txt, rtf, docx or doc), references, tables (including captions) and figure captions. Do not upload a PDF as your "Main Document".

- 2) A separate electronic file of each figure (EPS or print-quality PDF preferred (either format should be produced directly from original creation package), or original software format)
- 3) Included a 100 word media summary of your paper when requested at submission. Please ensure you have entered correct contact details (email, institution and telephone) in your user account
- 4) Included the raw data to support the claims made in your paper. You can either include your data as electronic supplementary material or upload to a repository and include the relevant doi within your manuscript
- 5) All supplementary materials accompanying an accepted article will be treated as in their final form. Note that the Royal Society will neither edit nor typeset supplementary material and it will be hosted as provided. Please ensure that the supplementary material includes the paper details where possible (authors, article title, journal name).

Best wishes,
Dr Laura Smith
Publishing Editor, Journals

RSC Associate Editor:

Comments to the Author:

I apologise that this has taken longer than usual. Reviewer 2 suggests moving some results to the Electronic Supplementary Material. Would you like to act on this?

RSC Subject Editor:

Comments to the Author:

(There are no comments.)

Reviewer comments to Author:

Reviewer: 1

Comments to the Author(s)

The authors have provided satisfactory modification to the manuscript by incorporating the raised comments. I appreciate authors effort and recommend it for publication.

Reviewer: 2

Comments to the Author(s)

Paper has been improved after major revision. I recommend it to be published in journal . I suggest to remove some results to supporting material, for example, Table1. please consider it.

Author's Response to Decision Letter for (RSOS-191590.R1)

See Appendices C & D.

Decision letter (RSOS-191590.R2)

30-Jan-2020

Dear Dr Kasim:

Title: Comparative study on photocatalytic activity of transition metals (Ag and Ni doped ZnO) nanomaterials synthesised via sol-gel method
Manuscript ID: RSOS-191590.R2

It is a pleasure to accept your manuscript in its current form for publication in Royal Society Open Science. The chemistry content of Royal Society Open Science is published in collaboration with the Royal Society of Chemistry.

RSC Associate Editor
Comments to the Author:
The manuscript can now be accepted.

Reviewer(s)' Comments to Author:

Appendix A

REVIEWER 1

The authors prepared Ag- and Ni/ZnO photocatalyst nanostructures, and studied their photocatalytic activity. The authors did a series of characterizations for the modified material. This work is somewhat interesting and worthy to be published in Royal Society Open Science; however, the following comments need to be addressed:

1. The title is inappropriate: is it '(Ag- and Ni-doped ZnO)' ?

Thank you for notifying on the title matter because such comment from reviewer is necessary to avoid misunderstanding. Therefore, the title used for this manuscript is considerable as it being compared with previous manuscript which has slight similarity in terms of the structured sentences. The “-” sign is removed and edited in latest revised manuscript.

References:

- 1) Yakout, S. M., & El-Sayed, A. M. (2019). Enhanced ferromagnetic and photocatalytic properties in **Mn or Fe doped** p-CuO/n-ZnO nanocomposites. *Advanced Powder Technology*. doi:10.1016/j.appt.2019.08.033
- 2) Türkyılmaz, Ş. Ş., Güy, N., & Özacar, M. (2017). Photocatalytic efficiencies of **Ni, Mn, Fe and Ag doped ZnO** nanostructures synthesized by hydrothermal method: The synergistic/antagonistic effect between ZnO and metals. *Journal of Photochemistry and Photobiology A: Chemistry*, 341, 39–50. doi:10.1016/j.jphotochem.2017.03.027
- 3) Ben Ali, M., Barka-Bouaifel, F., Sieber, B., Elhouichet, H., Addad, A., Boussekey, L., ... Boukherroub, R. (2016). Preparation and characterization of **Ni-doped ZnO–SnO₂** nanocomposites: Application in photocatalysis. *Superlattices and Microstructures*, 91, 225–237. doi:10.1016/j.spmi.2016.01.014

2. From the introduction, the significance of the work is insufficiency. There are many strategies to enhance the photocatalytic activity of ZnO. Particularly, the latest references about the ZnO or its hybrids are suggested to be added for the real value of general readers, as recommended below: Journal of physics and chemistry of solids, 106, (2017): 1-9; Journal of physics d-applied physics, 51(7), (2018): 075501. Thus, it can be clearly and specifically pointed out the novelty of this work in the introduction.

Thank you for the suggestion and recommendation. The correction has been made in introduction paragraph 3 (**line 44 to 54**) in the manuscript.

3. The quality of figures is too poor. Figures 1 and 2 should be merged. In Figures 3 and 4, the atomic % is misunderstanding: 0% O element? Figures 5-9 need to be revised.

Thank you for the comment. The resolution of figures has been revised, figures 1 and 2 have been merged accordingly.

O element is fundamentally assumed to be 100 %. It is known that EDS has limitation to quantify light element and O is one of them. EDS can detect the presence of O but to quantify them is tricky. Thus, EDS is good for quantifying the element as long as we do not quantify the O element. This is the reason why in this work, we just quantify element of Zn and (Ag or Ni) for accurate quantification of the element.

Figures 5 to 9 has been revised and the quality has been improved.

4. The mechanism of photocatalytic activity of Ag- and Ni/ZnO nanomaterials should be discussed.

Thank you for the suggestion. The mechanism has been added (**equation 6 to 14**) in the manuscript.

References:

- 1) Türkyılmaz, Ş. Ş., Güy, N., & Özacar, M. (2017). Photocatalytic efficiencies of Ni, Mn, Fe and Ag doped ZnO nanostructures synthesized by hydrothermal method: The synergistic/antagonistic effect between ZnO and metals. *Journal of Photochemistry and Photobiology A: Chemistry*, 341, 39–50. <https://doi.org/10.1016/j.jphotochem.2017.03.027>
- 2) Liu, Y., Zhang, Q., Xu, M., Yuan, H., Chen, Y., Zhang, J., ... You, B. (2019). Novel and efficient synthesis of Ag-ZnO nanoparticles for the sunlight-induced photocatalytic degradation. *Applied Surface Science*, 476, 632–640. <https://doi.org/10.1016/j.apsusc.2019.01.137>

- 3) Lee, K. M., Lai, C. W., Ngai, K. S., & Juan, J. C. (2016, January 1). Recent developments of zinc oxide based photocatalyst in water treatment technology: A review. *Water Research*. Elsevier Ltd. <https://doi.org/10.1016/j.watres.2015.09.045>
- 4) Ong, C. B., Ng, L. Y., & Mohammad, A. W. (2018). A review of ZnO nanoparticles as solar photocatalysts: Synthesis, mechanisms and applications. *Renewable and Sustainable Energy Reviews*. Elsevier Ltd. <https://doi.org/10.1016/j.rser.2017.08.020>

5. The whole manuscript (especially abstract and introduction) needs to be improved.

Answer: Thank you for the comment. The abstract (**line 4 to 11**) and introduction part (**line 44 to 54**) have been overhauled based on the earlier comments. The whole manuscript has been revised in detail.

6. In this manuscript, a large number of grammar errors may be found, and some expressions used in manuscript are confusing. The authors should polish English carefully.

Answer: Thank you for the comment. The manuscript has been proofread.

Appendix B

REVIEWER 2

Authors prepared the Ag- and Ni/ZnO photocatalyst nanostructures by a sol-gel method and investigated the effects of size, morphology, band gap, surface area and the number of active sites on the properties of photocatalyst. However, some issues exist in paper. I suggested revising it before being published.

1. In paper, there are little explanations of reason. For example, there are the effect of size, morphology, etc. Why?

Answer: Thank you for the comment. Little explanations on size and morphological study in this manuscript because there is not much differences between undoped and doped ZnO. We have added discussion on changes in average length for Ag and Ni/ZnO in the manuscript. Theoretically in previous reported literature, size and morphology of catalyst sample indeed played a major role. However, in our research work we found that active sites has better and beneficial role and somehow it takes vital part in influencing photocatalysis.

In theory, small size would give high surface area for catalyst. Plus, rod-like shape would be an added value to make it better. These factors would enhance the performance of nanocatalyst for the photocatalysis work. However, as mentioned in the manuscript for this research work, high surface area doesn't show much response for photocatalytic activity of azo dye compound. We managed to experiment the active sites study for 10 % of pure ZnO, Ag/ZnO and Ni/ZnO and it is believed to be a decisive factor in this research work.

References:

- 1) Umar, A., Kumar, R., Kumar, G., Algarni, H., & Kim, S. H. (2015). Effect of annealing temperature on the properties and photocatalytic efficiencies of ZnO nanoparticles. *Journal of Alloys and Compounds*, 648, 46–52. <https://doi.org/10.1016/j.jallcom.2015.04.236>
- 2) He, L., Tong, Z., Wang, Z., Chen, M., Huang, N., & Zhang, W. (2018). Effects of calcination temperature and heating rate on the photocatalytic properties of ZnO prepared by pyrolysis. *Journal of Colloid and Interface Science*, 509, 448–456. <https://doi.org/10.1016/j.jcis.2017.09.021>

2. Draft writing should be formatted and meet the requirement of journal. For example, paper abstract and characterization are required.

Answer: Thank you for the suggestion. Draft writing has been improved based on your suggestion and sectioned according to template format from The Royal Society Publishing.

3. Labels unit at x-/y axis should be marked in Figure 1, 2....., the font size in Figure 9 is not consistent with other Figures

Answer: Thank you for the notification. Labels unit in figures 1 and 2 showed the intensity with marked arbitrary unit. Therefore, the unit on y-axis is omitted. The font size in figure 9 has been revised and edited in the latest manuscript.

4. Please check the spelling of words.

Answer: Thank you for the comment. The manuscript has been proofread.

Appendix C

REVIEWER 1

The authors have provided satisfactory modification to the manuscript by incorporating the raised comments. I appreciate authors effort and recommend it for publication.

Thank you for all the comments and suggestions in previous session. Thank you again for reviewer's time investment and commitment throughout the process.

Appendix D

REVIEWER 2

Paper has been improved after major revision. I recommend it to be published in journal . I suggest to remove some results to supporting material, for example, Table1. please consider it.

Thank you for the suggestion. Table 1 has been placed under supporting material (currently named Table S1).

Thank you for all the comments and suggestions in previous session. Thank you again for reviewer's time investment and commitment throughout this process.